# NATURAL GALORE: ACCELERATING GALORE FOR MEMORY-EFFICIENT LLM TRAINING AND FINE-TUNING

## ABSTRACT

Training LLMs presents significant memory challenges due to growing size of data, weights, and optimizer states. Techniques such as data and model parallelism, gradient checkpointing, and offloading strategies address this issue but are often infeasible due to hardware constraints. To mitigate memory usage, alternative methods like Parameter-Efficient-Fine-Tuning (PEFT) and GaLore approximate weights or optimizer states. PEFT methods, such as LoRA, have gained popularity for fine-tuning LLMs, though they require a full-rank warm start. In contrast, GaLore allows full-parameter learning while being more memory-efficient. This work introduces *Natural GaLore*, a simple drop in replacement for AdamW, which efficiently applies the inverse Empirical Fisher Information Matrix to low-rank gradients using Woodbury's Identity. We demonstrate that incorporating second-order information speeds up optimization significantly, especially when the iteration budget is limited. Empirical pretraining on 60M, 130M, 350M, and 1.1B parameter Llama models on C4 data demonstrate significantly lower perplexity over GaLore without additional memory overhead. By fine-tuning RoBERTa on the GLUE benchmark using *Natural GaLore*, we demonstrate significant reduction in gap 86.05% vs 86.28% for full-finetuning. Furthermore, fine-tuning the TinyLlama 1.1B model for function calling using the TinyAgent framework shows that *Natural GaLore* achieving 83.09% accuracy on the TinyAgent dataset, significantly outperforms 16-bit LoRA at 80.06% and even surpasses GPT4-Turbo by 4%, all while using 30% less memory.

## 1 INTRODUCTION

Large Language Models (LLMs) have achieved remarkable performance across various disciplines, including conversational AI and language translation. However, training and fine-tuning these models demand enormous computational resources and are highly memory-intensive. This substantial memory requirement arises from storing billions of trainable parameters along with associated gradients and optimizer states.

To quantify this, consider a model with $\Psi$ parameters which is being trained using the Adam optimizer. In this case, storing parameters and their gradients in 16-bit precision formats like FP16 or BF16 requires $2\Psi$ bytes each. The associated optimizer states are typically stored in 32-bit precision (FP32) for numerical stability, necessitating an additional $4\Psi$ bytes for each parameter, gradient momentum, and variance, amounting to $12\Psi$ bytes. Therefore, the total memory requirement sums up to $16\Psi$ bytes. When accounting for model-dependent memory, such as activations during forward and backward passes, and residual memory, like temporary buffers and memory fragmentation, the overall memory footprint can easily exceed $18\Psi$ bytes (Raffel et al., 2020; Touvron et al., 2023; Chowdhery et al., 2022).

This enormous memory demand poses significant challenges, especially when training LLMs on hardware with limited memory capacity. As models continue to scale, efficient memory utilization becomes critical for making training feasible and accessible. In this work, we develop an efficient adaptation to the GaLore algorithm (Zhao et al., 2024a), which significantly reduces the memory footprint during training and fine-tuning of LLMs by approximating the optimizer state. Our approach, *Natural GaLore*, leverages the low-rank structure of gradients and incorporates second-order information to achieve faster convergence and higher performance without additional memory over-

head and can be used as a drop in replacement [1] to standard optimization algorithms like Adam and AdamW.

**Parallel and Distributed Training Techniques**   Researchers have developed various distributed computing techniques that leverage system-level optimizations and hardware resources to mitigate the substantial memory requirements in training LLMs.

One prominent framework is *Distributed Data-Parallel (DDP)* that combines data parallelism where the training dataset is partitioned across multiple devices or nodes, with efficient gradient synchronization mechanisms, minimizing communication overhead. While data parallelism efficiently utilizes multiple GPUs, it can still face memory bottlenecks when model sizes exceed the memory capacity of a single device.

*Model parallelism* addresses this limitation by partitioning the model across multiple devices, allowing for the training of models that are too large to fit into the memory of a single GPU. Techniques like *pipeline parallelism* (Huang et al., 2019) and *tensor parallelism* (Shoeybi et al., 2019) enables the distribution of different layers or partitions of layers across devices. However, model parallelism introduces communication overhead and can be complex to implement effectively.

Another effective technique is *gradient checkpointing* (Chen et al., 2016), which reduces memory usage by selectively storing only a subset of activations during the forward pass and recomputing them during the backward pass as needed. This approach trades increased computational overhead for reduced memory consumption, enabling the training of deeper models without exceeding memory constraints.

*Memory offloading* strategies, such as those implemented in ZeRO-Offload (Rajbhandari et al., 2020), move optimizer states and gradients to CPU memory when not actively in use, freeing up GPU memory for other operations. ZERO can also partition optimizer states and gradients across DDP processes, eliminating redundancy and significantly reducing memory footprint. *Fully Sharded Data Parallel* (Zhao et al., 2020) extends this concept by sharding model parameters in addition to optimizer states and gradients.

These system-level optimizations have been instrumental in training state-of-the-art LLMs such as LLaMA3 (Touvron et al., 2023), GPT-3 (Brown et al., 2020), Mistral (Jiang et al., 2023), and Gopher (Rae et al., 2021) on multi-node, multi-GPU clusters.

While these distributed computing solutions enable the training of large models by leveraging extensive hardware resources, they come with increased system complexity and operational costs. Therefore, there is a pressing need for alternative approaches that reduce memory consumption without relying solely on distributed computing resources. Optimization techniques that approximate parameters or optimizer states offer a promising direction for making LLM training more accessible and efficient.

**Parameter-Efficient Fine-Tuning**   PEFT techniques efficiently adapt pre-trained language models to various downstream applications without fine-tuning all the model's parameters (Ding et al., 2022), significantly reducing the computational and memory overhead.

Among these techniques, the popular LoRA (Hu et al., 2022) parametrizes a weight matrix $W \in \mathbb{R}^{n \times m}$ as:

$$W = W_0 + BA, \tag{1}$$

where $W_0$ is a frozen full-rank pre-trained weight matrix, and $B \in \mathbb{R}^{n \times r}$ and $A \in \mathbb{R}^{r \times m}$ are trainable low-rank adapters to be learned during fine-tuning. Since the rank $r \ll \min(m, n)$, the adapters $B$ and $A$ contain significantly fewer trainable parameters, reducing memory requirements for both parameter and optimizer states.

LoRA has been extensively used to reduce memory usage during fine-tuning, effectively enabling large models to be adapted to new tasks with minimal additional memory overhead. There are a few variants of LoRA proposed to enhance its performance (Renduchintala et al., 2023; Sheng et al., 2023; Zhang et al., 2023; Xia et al., 2024), supporting multi-task learning (Wang et al., 2023), and

---

[1]All code to reproduce the results are provided in the supplementary

further reducing the memory footprint (Dettmers et al., 2023). Its variant, ReLoRA (Lialin & Schatz, 2023), extends LoRA's approach to pre-training by periodically updating the frozen weight matrix $W_0$ using the previously learned low-rank adapters. This incremental updating allows for continual learning without storing entire optimizer states for all parameters, leading to faster training times and lower computational costs. Furthermore, this allows for rapid adaptation of large models to multiple downstream tasks without storing separate copies of the entire model for each task.

Despite their benefits, recent works have highlighted several limitations of low-rank reparameterization approaches. LoRA does not consistently achieve performance comparable to full-rank fine-tuning, particularly in complex tasks (Xia et al., 2024). In pre-training from scratch, methods like ReLoRA require an initial phase of full-rank model training as a warmup before optimizing in the low-rank subspace (Lialin & Schatz, 2023). The shortcomings of low-rank parameter reparameterization suggest that alternative strategies are needed to achieve both memory efficiency and high performance.

**Gradient Low-Rank Projection (GaLore)** An alternative to parameter approximation is the approximation of the optimizer states. By reducing the memory footprint associated with optimizer states, it is possible to maintain full-parameter learning—thus preserving model capacity and performance—while achieving significant memory savings.

The core idea behind GaLore (Zhao et al., 2024a) is to exploit the slowly changing low-rank structure of the gradient matrix $g \in \mathbb{R}^{n \times m}$, rather than approximating the weights. During neural network training, gradients naturally exhibit low-rank properties, a phenomenon studied extensively in both theoretical and practical settings (Zhao et al., 2022; Cosson et al., 2023; Yang et al., 2023). This intrinsic low-rank structure of gradients has been applied to reduce communication costs (Wang et al., 2018; Vogels et al., 2020) and to decrease memory footprints during training (Gooneratne et al., 2020; Zhao et al., 2024b).

Specifically, consider the compact SVD decomposition of the gradient matrix $\mathbf{g} = \mathbf{P}\Sigma\mathbf{Q}^T$, where $\mathbf{P} \in \mathbb{R}^{n \times r}$ and $\mathbf{Q} \in \mathbb{R}^{m \times r}$ are the associated semi-orthognal matrices. Then, GaLore projects the gradient matrix $\mathbf{g}$ into a low-rank form:

$$\mathbf{g}_{\text{low-rank}} = \mathbf{P}^T\mathbf{g}. \tag{2}$$

Here, $r \ll \min(n, m)$ is the target rank, $n$ is the parameter count, $m$ is the batch size and $\mathbf{g}_{\text{low-rank}}$ serves as an efficient approximation of the original gradient. The projection matrix $\mathbf{P}$ is updated periodically (e.g., every 200 iterations), which incurs minimal amortized computational cost.

By operating on low-rank approximations of the gradients, GaLore significantly reduces the memory footprint, leading to up to **30%** memory reduction compared LoRA (Zhao et al., 2024a). Moreover, GaLore maintains full-parameter learning, allowing updates to all model parameters, leading to better generalization and performance than low-rank adaptation methods. Further, GaLore is agnostic to the choice of optimizer and can be easily integrated into existing optimization algorithms with minimal code modifications.

While GaLore offers significant memory savings and enables full-parameter learning, its performance has yet to match that of optimizers in full optimizer state space. Reliance on low-rank gradient approximations may not fully capture the rich optimization dynamics. These limitations suggest that while GaLore is a valuable step toward memory-efficient training, further enhancements are necessary to bridge the performance gap with standard optimizers.

**Our Approach** In this work, we propose to bridge the gap by incorporating a second-order regularizer into the low-rank gradient estimate, which adjusts parameter updates more effectively, leading to faster convergence. We show that applying the inverse of the empirical Fisher Information Matrix (FIM) to the low-rank gradients leads to variance reduction of the gradient estimate, incorporates information about the curvature of the loss landscape, and reduces dependence on the starting point. All of these lead to significantly faster convergence, especially in a limited iteration regime.

We introduce the *Natural GaLore* algorithm, a matrix-free algorithm for efficiently applying the inverse FIM to the low-rank gradients, using Woodbury Identity, Cholesky Decomposition, and Matrix-Vector Products, all of which can be efficiently implemented on the GPU. Further, our approach does not require any explicit layer-wise information or significant computational overhead, as is seen in existing approaches like K-Fac (Martens & Grosse, 2015).

We validate the effectiveness of *Natural GaLore* through extensive empirical evaluations. Pre-training experiments on LLaMA models with 60M, 300M, and 1.1B parameters using the C4 dataset demonstrate that *Natural GaLore* achieves significantly lower perplexity than GaLore without additional memory overhead, indicating faster convergence within the same computational budget.

Furthermore, we showcase the practical benefits of *Natural GaLore* in fine-tuning tasks. We fine-tune the TinyLlama 1.1B model for function calling using the TinyAgent framework. Our results show that *Natural GaLore* significantly outperforms LoRA in this setting, achieving an accuracy of **83.09%** on the TinyAgent dataset. This performance significantly surpasses 16-bit LoRA and exceeds that of GPT-4-turbo by 4%, all while using **30%** less memory.

## 2 ACCELERATING GALORE WITH NATURAL GRADIENTS

### 2.1 NEXT TOKEN PREDICTION

Generative LLMs are trained to predict the next token in a sequence based solely on the previously observed tokens. This "causal" approach respects the temporal order of language, ensuring that the model's predictions at any point depend only on past and not future inputs.

Given a sequence of tokens $x = (x_1, x_2, \ldots, x_T)$, the objective is to maximize the likelihood of a sequence by decomposing it into a product of conditional probabilities:

$$\text{Prob}_\theta(x) = \prod_{t=1}^{T} \text{Prob}_\theta(x_t \mid x_{<t}) \tag{3}$$

where $x_{<t} = (x_1, x_2, \ldots, x_{t-1})$ represents all tokens before position $t$ and $\text{Prob}_\theta(x_t \mid x_{<t})$ is the probability of the next token given all previous tokens and the parameter $\theta \in \mathbb{R}^{n \times m}$.

This is equivalent to minimizing the Negative Log-Likelihood (NLL) of the observed sequences, which is the cross-entropy loss between the predicted probability distribution and the actual next token:

$$\Phi(\theta) = -\sum_{t=1}^{T} \log \text{Prob}_\theta(x_t \mid x_{<t}) \tag{4}$$

This loss penalizes the model more when it assigns lower probabilities to the correct next token. By minimizing this loss, the model learns to assign higher probabilities to appropriate continuations of text. However, the loss is non-convex and high-dimensional, for LLMs the dataset is also massive, making the optimization problem very challenging.

### 2.2 LOW-RANK GRADIENT DESCENT

Stochastic gradient descent algorithms are iterative, where each step aims to find the optimal update direction that minimizes the loss function locally. Now in the case of GaLore, the update direction is restricted to the affine subspace $\mathbf{u}_k \in \theta_k + \text{Range}(\mathbf{P}_k)$. Here $\mathbf{P}_k \in \mathbb{R}^{n \times r}$ is the left projection matrix, calculated using the compact SVD decomposition of the gradient matrix $\nabla_\theta \Phi(\theta_k) = \mathbf{P}_k \Sigma \mathbf{Q}_k^T$.

Then, the local neighborhood around this update can be defined using the Taylor series expansion (Lin et al., 2022):

$$\Phi(\theta_k + \mathbf{P}_k \mathbf{u}_k) \approx \Phi(\theta_k) + \mathbf{g}_k^T \mathbf{u}_k + \frac{1}{2} \mathbf{u}_k^T \mathbf{H}_k \mathbf{u}_k \tag{5}$$

where $\mathbf{g}_k = \mathbf{P}_k^T \nabla_\theta \Phi(\theta_k)$ is the low rank projected gradient and $\mathbf{H_k} = \mathbf{P}_k^T \nabla_\theta^2 \Phi(\theta) \mathbf{P}_k$ is the Hessian matrix.

However, the Hessian matrix $\mathbf{H}_k$ is often computationally expensive to compute and store, especially for large-scale language models (LLMs) with billions of parameters. Fortunately, precisely under

the condition that the loss function can be represented in terms of KL divergence between the actual and approximated distributions [4], then $\mathbf{H_k}$ can be approximated by the FIM. The FIM is defined as the expectation of the Hessian of the negative log-likelihood w.r.t. the data distribution:

$$\mathbf{F}_k = \mathbb{E}_{x \sim p_{\text{data}}}[\mathbf{H}_k] \tag{6}$$

The FIM captures the curvature of the loss landscape and provides a natural metric for the optimization process. Hence, it can better adjust parameter updates according to the geometry of the parameter space. However, as the theoretical data distribution is unknown, in practice, we need to estimate it using the empirical FIM (Martens, 2014) defined by:

$$\hat{\mathbf{F}}_k = \frac{1}{h} \sum_{k=1}^{h} \mathbf{g_k}\mathbf{g_k}^T \tag{7}$$

where $h$ is the history of gradients from past batches we would like to consider. Then, the optimal direction $\mathbf{u}_k^*$, which minimizes the loss in this local neighborhood, is given by (cite Fuji et al. paper):

$$\mathbf{u}_k^* = \hat{\mathbf{F}}_k^{-1}\mathbf{g}_k \tag{8}$$

This leads to the optimal gradient descent update step:

$$\theta_{k+1} = \theta_k - \eta\mathbf{P}_k\mathbf{u}_k^* \tag{9}$$

for some learning rate $\eta$.

Many popular stochastic optimization algorithms approximate the diagonal of the empirical FIM using second-moment estimates of the gradient $\mathbf{g}_k$, which when added with Polyak style parameter averaging (i.e., momentum), asymptotically achieve the optimal Fisher efficient convergence rate (Martens, 2020).

For instance, in the case of Adam (Kingma & Ba, 2014), the optimal update step is approximated by including the momentum term $\mathbf{m}_k \in \mathbb{R}^{r \times m}$ and the learning rate $\eta$ is scaled by the square root of the second moment estimate $\mathbf{v}_k \in \mathbb{R}^{r \times m}$. With all operations being elementwise, the update direction becomes:

$$\mathbf{m}_k = \beta_1\mathbf{m}_{k-1} + (1 - \beta_1)\mathbf{g}_k \tag{10}$$

$$\mathbf{v}_k = \beta_2\mathbf{v}_{k-1} + (1 - \beta_2)\mathbf{g}_k^2 \tag{11}$$

$$\mathbf{u_k}^* = \mathbf{m}_k/\sqrt{\mathbf{v}_k + \epsilon} \tag{12}$$

This update, when applied to [9], gives the GaLore optimization algorithm, which is memory efficient as it only requires storing the projection matrix and the costly optimizer states $(g_k, m_k, v_k)$ are now significantly reduced by a factor of $\frac{n}{r}$, where the rank $r$, can be chosen based on the tradeoff between memory limitations and performance requirements.

## 2.3 NATURAL GALORE AND FISHER EFFICIENCY

Despite clear advantages, the performance of GaLore is not on par with AdamW (Loshchilov & Hutter, 2017) optimization on the original space. To bridge this gap, we propose *Natural GaLore*, which uses the full empirical FIM, thereby incorporating the missing second-order interaction information in the optimization process.

As we now argue, this leads to a much more favorable dependence on the starting point, which means that the optimizer can make much more progress given a limited iteration budget. Further, when using a decaying learning rate schedule like with AdamW (Loshchilov & Hutter, 2017), the asymptotic convergence rate can be faster (Martens, 2020) by a significantly large constant factor.

Natural gradient descent is known (Martens, 2020) to be Fisher efficient, precisely for our loss function [4]. Fisher efficiency means that the natural gradient estimator asymptotically achieves the lowest possible variance among all unbiased gradient estimators.

For *Natural GaLore*, the gradient descent update [9] leads to a sequence of estimates $\theta_k$ whose variance satisfies (Amari, 1998):

$$\text{Var}[\theta_k] = \frac{1}{mk}\mathbf{F}_k^{-1}(\theta_k^*) + \mathcal{O}\left(\frac{1}{k^2}\right) \tag{13}$$

which is asymptotically the smallest possible variance matrix satisfying the Cramér-Rao lower bound, that any unbiased estimator computed from $mk$ training samples can have, with $m$ being the batch size.

Here, $\theta_k^*$ is the local optimum in the neighborhood defined by the Taylor series expansion [5] around the update direction. This is an important caveat, as the guarantee is only for local convergence in a convex neighborhood. The loss function is non-convex, so the property can not be stated to hold for the global optimum.

The result also relies on the computation of the exact FIM $\mathbf{F}_k(\theta_k)$ using the entire data distribution, which is not practical. The Fisher efficiency guarantee is, however, only approximately satisfied when using the empirical FIM $\hat{\mathbf{F}}_k$ instead. Nevertheless, we still get a variance reduction in the gradient estimates, leading to faster convergence and better optimization performance in the early stages of training large-scale models, making it especially valuable for training with a limited iteration budget.

Further, incorporating second-order information through the empirical FIM allows the optimizer to account for the curvature of the loss landscape, enabling natural gradient descent to take more informed steps than standard gradient descent, potentially escaping flat regions or navigating steep ravines more effectively.

In (Martens, 2020), it was shown that the expected update direction can be expressed as a sum of two terms, one that scales as $\mathcal{O}(1/k)$, which is independent of the starting point and another that scales as $\mathcal{O}(1/k^2)$, which is dependent on the starting point. If momentum is applied to the gradient estimator, the first term becomes independent of the choice of FIM estimator, thereby not leading to any asymptotic improvements. However, regularizing with the empirical FIM estimate can significantly reduce the constant factor associated with the starting-point-dependent second term. This leads to practical performance gains in finite iteration regimes (although negligible for large $k$).

Finally, the Fisher efficiency result also assumes that the model can perfectly capture the data distribution, a condition known as *realizability*. However, with the growing size of LLMs, this assumption is likely to hold, thereby satisfying the conditions for the guarantee. Therefore, especially in low-resource settings, *Natural GaLore* can be a promising approach for training LLMs under memory constraints.

## 2.4 NATURAL GRADIENT TRANSFORM

Our *Natural GaLore* algorithm is designed to efficiently apply the inverse empirical FIM to low-rank gradients using Woodbury's Identity. Most of the steps in the algorithm are similar to GaLore (Zhao et al., 2024a), with the critical difference being the incorporation of the natural gradient transform.

In order to implement the natural gradient transform, we compute the inverse of the empirical FIM and apply it to the gradient $\mathbf{g_k}$ using Woodbury's Identity, which allows us to efficiently compute the inverse of a matrix of the form $A + UBU^T$. Woodbury's Identity states that:

$$(A + UBU^T)^{-1} = A^{-1} - A^{-1}U(B^{-1} + U^T A^{-1}U)^{-1}U^T A^{-1} \tag{14}$$

Now, if we choose $\hat{\mathbf{F}}_k = \lambda I + GG^T$, $A = \lambda I$, $U = G$, and $B = I$, where $G = [\text{vec}(\mathbf{g}_k), \text{vec}(\mathbf{g}_{k-1}), \dots, \text{vec}(\mathbf{g}_{k-s})]$ is the stacked gradient matrix over the past $s$ gradients and $\lambda$ is a small constant for Tikhonov regularization, then, the inverse of the empirical FIM applied to the gradient $\mathbf{g_k}$ i.e. the natural gradient $\tilde{\mathbf{g}}_k = \hat{\mathbf{F}}_k^{-1}\mathbf{g_k}$ can be calculated as:

$$\tilde{\mathbf{g}}_k = \frac{1}{\lambda}\mathbf{g_k} - \frac{1}{\lambda}G\left(\lambda I + G^T G\right)^{-1} G^T \mathbf{g_k} \tag{15}$$

To compute the above formula efficiently, let $S = I + \frac{1}{\lambda}G^T G \in \mathbb{R}^{s \times s}$ and $y = G^T \mathbf{g_k}$. Cholesky decomposition is used to solve for $z$ in

$$Sz = y \tag{16}$$

which requires only $\mathcal{O}(s^2)$ time. Then, the final natural gradient estimate can be computed using only matrix-vector products, which is very memory efficient:

$$\tilde{\mathbf{g}}_k = \frac{1}{\lambda}\mathbf{g_k} - \frac{1}{\lambda^2}Gz \tag{17}$$

This natural gradient estimate $\tilde{\mathbf{g}}_k$ can then be sent to the Adam optimizer [12], and the model parameters the same way as in GaLore.

## 3 EXPERIMENTS

We evaluate *Natural GaLore* on pre-training and fine-tuning tasks for LLMs. All experiments are conducted on a single node with 8 NVIDIA A100 GPUs to leverage high-performance computing capabilities, yet stay within reasonable limits.

### 3.1 PRE-TRAINING ON THE C4 DATASET

To assess the effectiveness of *Natural GaLore*, we apply it to pre-train LLaMA-based language models of sizes ranging from 60 million to 1.1 billion parameters, on the C4 dataset. The C4 dataset is a colossal, cleaned version of the Common Crawl Corpus, primarily intended for pre-training language models and word representations (Raffel et al., 2020). It provides a diverse and extensive corpus, making it suitable for evaluating pre-training methods in realistic scenarios.

We adopt the experimental setup from Lialin & Schatz (2023), utilizing a LLaMA-based[2] architecture with RMSNorm and SwiGLU activations (Shazeer, 2020; Touvron et al., 2023). We maintain the same set of hyperparameters for each model size across all methods, except for the learning rate, which is tuned individually to ensure optimal performance. All experiments use the BF16 format to reduce memory usage without compromising computational efficiency, the same computational budget and the best validation perplexity is reported.

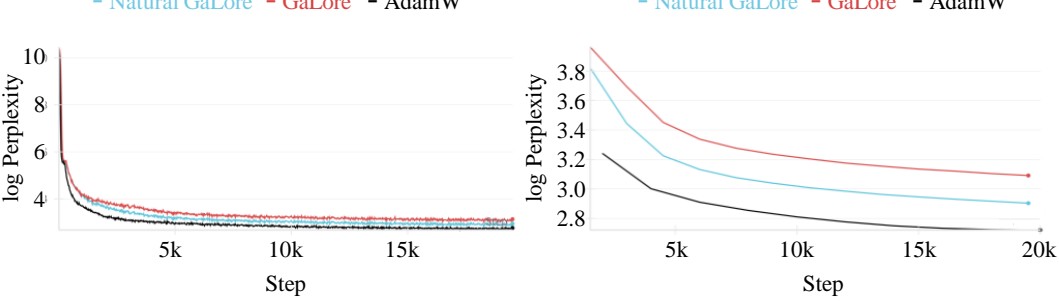

Figure 1: Training and Validation log Perplexity for Llama 1.1B

Table 1 presents the validation perplexity and memory consumption for models trained with different methods and Figure 1 shows the training run for the Llama 1.1B model. Our proposed *Natural GaLore* consistently outperforms GaLore (Zhao et al., 2024a) across all model sizes, achieving validation perplexities closer to the full-rank baseline while maintaining significant memory savings. Furthermore, *Natural GaLore* exhibits lower perplexities and greater memory consumption compared to other low-rank adaptation methods like LoRA and ReLoRA, due to their less efficient use of low-rank structures and the need for additional optimizer states.

### 3.2 FINE-TUNING ROBERTA-BASE ON THE GLUE BENCHMARK

To further evaluate the effectiveness of *Natural GaLore*, we conduct experiments on the General Language Understanding Evaluation (GLUE) benchmark using the pre-trained RoBERTa-Base

---

[2]LLaMA materials in our paper are subject to the LLaMA community license.

|  | 60M | 130M | 350M | 1.1B |
|---|---|---|---|---|
| Full-Rank | 3.52 (0.36G) | 3.22 (0.76G) | 2.93 (2.06G) | 2.72 (7.80G) |
| *Natural GaLore* | **3.53** (0.24G) | **3.22** (0.52G) | **2.93** (1.22G) | **2.80** (4.38G) |
| GaLore | 3.56 (0.24G) | 3.24 (0.52G) | 2.95 (1.22G) | 2.90 (4.38G) |
| Low-Rank | 4.35 (0.26G) | 3.82 (0.54G) | 3.62 (1.08G) | 4.96 (3.57G) |
| LoRA | 3.55 (0.36G) | 3.52 (0.80G) | 3.24 (1.76G) | 2.96 (6.17G) |
| ReLoRA | 3.61 (0.36G) | 3.38 (0.80G) | 3.37 (1.76G) | 2.91 (6.17G) |
| Rank $r/d_{\mathrm{model}}$ | 128 / 256 | 256 / 768 | 256 / 1024 | 512 / 2048 |
| Training Tokens | 1.1B | 2.2B | 6.4B | 13.1B |

Table 1: Comparison of *Natural GaLore* with other low-rank algorithms on pre-training various sizes of LLaMA models on the C4 dataset. Validation log perplexity is reported (averaged over 5 runs), along with a memory estimate (in gigabytes) of the total parameters and optimizer states based on BF16 format.

model. The GLUE benchmark is a collection of nine natural language understanding tasks, including single-sentence tasks like CoLA (Warstadt et al., 2019), similarity and paraphrase tasks like MRPC (Dolan & Brockett, 2005) and STS-B (Cer et al., 2017), and inference tasks like RTE (Dagan et al., 2006), MNLI (Williams et al., 2018), and QNLI (Rajpurkar et al., 2016). This benchmark is widely used to assess the performance of language models on diverse linguistic phenomena.

In our experiments, we fine-tune the RoBERTa-Base model using *Natural GaLore* and compare its performance with full fine-tuning and LoRA (Hu et al., 2022). We focus on memory-efficient fine-tuning methods to reduce the computational footprint while maintaining high performance. For each method, we report the average score across all GLUE tasks and individual task scores.

We use the same training hyperparameters across all methods for a fair comparison. The batch size is 32, and we fine-tuned each model for three epochs. The learning rate is selected from {1e-5, 2e-5, 3e-5} based on the best validation performance for each task. For *Natural GaLore* and LoRA, we experiment with rank values of 4 and 8 to study the trade-off between performance and memory efficiency.

Table 2 presents the results of our experiments. *Natural GaLore* consistently achieves comparable or better performance than LoRA across most tasks while using less memory. Precisely, with a rank of 4, *Natural GaLore* attains an average score of **86.05**, closely matching the complete fine-tuning baseline of 86.28 and outperforming LoRA's average score of 85.61. This demonstrates that *Natural GaLore* can effectively fine-tune large models with reduced memory consumption without sacrificing performance.

|  | Memory | CoLA | STS-B | MRPC | RTE | SST-2 | MNLI | QNLI | QQP | Avg |
|---|---|---|---|---|---|---|---|---|---|---|
| Full Fine-Tuning | 747M | 62.24 | 90.92 | 91.30 | 79.42 | 94.57 | 87.18 | 92.33 | 92.28 | 86.28 |
| *Natural GaLore* (**rank=4**) | 253M | 61.50 | **90.80** | **92.10** | **79.50** | **94.20** | **87.05** | **92.30** | 91.15 | **86.05** |
| GaLore (rank=4) | 253M | 60.35 | 90.73 | 92.25 | 79.42 | 94.04 | 87.00 | 92.24 | 91.06 | 85.89 |
| LoRA (rank=4) | 257M | **61.38** | 90.57 | 91.07 | 78.70 | 92.89 | 86.82 | 92.18 | **91.29** | 85.61 |
| *Natural GaLore* (**rank=8**) | 257M | 61.70 | **90.90** | **92.25** | **79.80** | **94.40** | **87.20** | **92.35** | 91.25 | **86.23** |
| GaLore (rank=8) | 257M | 60.06 | 90.82 | 92.01 | 79.78 | 94.38 | 87.17 | 92.20 | 91.11 | 85.94 |
| LoRA (rank=8) | 264M | **61.83** | 90.80 | 91.90 | 79.06 | 93.46 | 86.94 | 92.25 | 91.22 | 85.93 |

Table 2: Evaluating *Natural GaLore* for memory-efficient fine-tuning on the GLUE benchmark using pre-trained RoBERTa-Base. We report the average score of all tasks. Memory consumption is reported in millions of parameters (M).

### 3.3 Fine-Tuning TinyLlama 1.1B for Function Calling in Advanced Agentic Systems

Advanced Agentic Systems (AAS) require language models that can understand and generate code snippets to integrate various tools and APIs, fulfilling user queries through function-calling. We utilize the TinyAgent framework, which provides an end-to-end pipeline for training and deploying task-specific LLM agents capable of efficient and accurate function-calling (Erdogan et al., 2024) to drive agentic systems at the edge.

Given a natural language query, the LLM agent must generate a sequence of pre-defined function-calls that accomplish the desired tasks. The challenge lies in determining the appropriate arguments, to call the correct functions, in the right order while respecting interdependencies among the functions.

LLMCompiler Kim et al. (2023), is a framework that enables language models to perform function-calling by first generating a function-calling plan, which includes the required functions and arguments. The LLMCompiler then compiles this plan into an executable sequence of function-calls. The critical aspect is training the model to produce a function-calling plan with the correct syntax and dependencies.

The off-the-shelf pre-trained TinyLlama 1.1B (instruct-32k) model performs poorly on this task. The model generates incorrect sets of functions, hallucinated function names, fails to respect dependencies, and passes arguments incorrectly. This underperformance is expected, as the model was initially trained on datasets like SlimPajama and StarCoder, which are not specific to function-calling tasks. To address this, we follow the TinyAgent framework (Erdogan et al., 2024) and fine-tune the TinyLlama 1.1B model on a high-quality, curated dataset designed for function-calling.

**TinyAgent Dataset**  The TinyAgent dataset (Erdogan et al., 2024) is a meticulously curated collection aimed at building a local agentic system for function-calling on Apple MacBooks for day-to-day tasks. It contains 40K examples of natural language queries and corresponding function-calling plans. The dataset is divided into 38K training examples, 1K validation examples, and 1K test examples. It encompasses 16 tasks, including Email, Contacts, SMS, Calendar, Notes, Reminders, File Management and Zoom Meetings. Each task has predefined scripts that the model needs to generate. The dataset is intentionally challenging, requiring the model to understand dependencies between function-calls and the arguments to be passed.

**Fine-Tuning Procedure**  We fine-tune the TinyLlama 1.1B model on the TinyAgent dataset for three epochs using a batch size of 32. The learning rate is set to $7 \times 10^{-5}$. After each epoch, the model is evaluated on the validation set, and the best-performing model is selected based on validation performance to be evaluated on the test set.

During fine-tuning, the prompt includes descriptions of the ground truth functions and irrelevant functions serving as negative samples. This strategy encourages the model to learn to select the correct functions rather than merely memorizing the ground truth. Additionally, several in-context examples demonstrate how queries are translated into function-calling plans. These examples are selected using a Retrieval-Augmented Generation (RAG) process based on the user's query from the training data and a DeBERTa-v3-small model (He et al., 2021) fine-tuned for multi-label classification for retrieval among the 16 tools.

The training objective is then to maximize the accuracy of the generated function-calling plans. Success is defined by the model generating the correct plan with the proper set of function-calls, correct arguments, and the appropriate order of function-calls. Verifying the selection of the correct set of functions involves straightforward set comparison. However, ensuring the correctness of arguments and the order of function-calls is more complex and requires constructing the associated Directed Acyclic Graph to check for equality.

| Model | Weight Precision | Latency (seconds) | Model Size (GB) | Success Rate (%) |
|---|---|---|---|---|
| GPT-3.5 | Unknown | 3.2 | Unknown | 65.04 |
| GPT-4-Turbo | Unknown | 3.9 | Unknown | 79.08 |
| TinyAgent-1.1B | 16-bit (*Natural GaLore*) | 3.9 | 2.2 | **83.09** |
|  | 16-bit (LoRA) | 3.9 | 2.2 | 80.06 |
| TinyAgent-7B | 16-bit (Erdogan et al., 2024) | 19.5 | 14.5 | 84.95 |

Table 3: Latency, size, and success rate of TinyAgent models before and after quantization. Latency is the end-to-end latency of the function calling planner, including the prompt processing time and generation.

**Results and Discussion**    After fine-tuning, the TinyLlama 1.1B model's success rate on the test set improved significantly. Table 3 presents the latency, model size, and success rate of various models on the TinyAgent dataset. As shown, *Natural GaLore* improves the success rate of the 1.1B model from 80.06% (16-bit LoRA) to **83.09%**, also surpassing GPT-4-Turbo by 4% and approaching the performance of the larger TinyAgent-7B model, which achieves 84.95%.

These results demonstrate that *Natural GaLore* not only enhances the performance of smaller models like the 1.1B parameter TinyLlama but also makes them competitive with significantly larger models. By efficiently incorporating second-order information through low-rank natural gradient updates, *Natural GaLore* enables smaller models to achieve higher accuracy without additional memory overhead.

## 4 Conclusion

We have introduced *Natural GaLore*, a memory-efficient pre-training and fine-tuning strategy for large language models. *Natural GaLore* significantly reduces memory usage—by up to 65.5% in optimizer states—while maintaining or even improving performance in large-scale LLM pre-training and fine-tuning tasks. By incorporating second-order information through an efficient approximation of the inverse Empirical Fisher Information Matrix, *Natural GaLore* enhances convergence rates, especially in regimes with a limited iteration budget.

Importantly, *Natural GaLore* can serve as a *drop-in replacement* for standard optimizers like AdamW and integrates seamlessly into existing training pipelines. Our experimental results highlight the *reproducibility* and effectiveness of *Natural GaLore* across various tasks, including pre-training LLaMA models and fine-tuning on the GLUE benchmark, as well as the TinyAgent function calling tasks. This makes it a compelling choice for large-scale pre-training scenarios where both memory efficiency and model performance are critical.

In the future we want to explore (1) further enhancing memory efficiency by employing low-memory and structured projection matrices, and (2) more extensive empirical evaluation on fine-tuning AAS on a wide variety of tasks. We also hope that our work will inspire future research on memory-efficient training methods from the perspective of optimizer state approximation. We believe that *Natural GaLore* will be a valuable tool for the community, enabling the training of large-scale models on consumer-grade hardware with limited resources.

## Impact Statement

This work aims to improve the memory efficiency of training LLMs, thereby reducing the environmental impact of LLM pre-training and fine-tuning. By enabling the training of larger models on hardware with lower memory requirements, our approach helps to minimize energy consumption and carbon footprint associated with training LLMs. Furthermore, by making advanced model training more accessible, we contribute to democratizing AI research and development, allowing a broader community to engage with large-scale models without the need for expensive computational resources.

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
