# OpenReview forum: "Natural GaLore: Accelerating GaLore for memory-efficient LLM Training and Fine-tuning"
_ICLR.cc/2025/Conference — ICLR 2025 Conference Withdrawn Submission_

### Official Review · Reviewer_FApJ · 2024-10-22

**Soundness:** 2
**Presentation:** 3
**Contribution:** 1
**Rating:** 3
**Confidence:** 3

**Summary:**

This paper builds upon GaLore by incorporating second-order terms and using the Fisher Information Matrix to estimate the computationally expensive Hessian matrix for acceleration. Experiments show that this work achieves improved performance compared to GaLore without increasing memory consumption.

**Strengths:**

1. The authors provide a detailed mathematical derivation process.
2. The authors offer a comprehensive introduction to memory efficiency improvements in LLM training.

**Weaknesses:**

1. This work merely adds second-order terms to GaLore, lacking sufficient innovation.
2. The performance improvement over GaLore is not significant, and it introduces additional time consumption. However, the authors did not specify the extent of the increase in time consumption, raising concerns about the practicality of this work.

**Questions:**

1. In Section 2.4, it is mentioned that G is the stacked gradient matrix over the past s gradients. Would storing past gradients increase memory consumption?
2. How much additional time does this work consume compared to GaLore? Could GaLore train more steps or epochs within the same time, thereby achieving better performance than this work?

---

### Official Review · Reviewer_obMc · 2024-10-25

**Soundness:** 4
**Presentation:** 3
**Contribution:** 2
**Rating:** 6
**Confidence:** 3

**Summary:**

This paper proposes a memory-efficient training method for LLMs, named Natural GaLore. Compared with previous work, GaLore, this paper applies second-order information through inverse Empirical Fisher Information Matrix to reach better model convergence. There is no memory overhead over GaLore for this method.

**Strengths:**

-Efficient training is important for LLMs. How to accelerate model convergence with less memory is a hot topic for both academic and industrial circles.

-This paper proposes an insightful method to accelerate convergence over baseline GaLore. Extensive experiments are shown.

-The paper is easy to follow.

**Weaknesses:**

-My major concern is the experimental improvement of Natural GaLore over the baseline Galore. For example, in Table 1, GaLore is a baseline close to full-rank one, so the improvement of Natural GaLore over it seems to be limited.

-Training throughputs of different methods should be compared, as it is an important metric used to be traded-off for memory efficiency.

-Results of different runs (e.g. Table 1) should be reported to prove the method reduces dependence on the starting points, which is one of the main advantages of the proposed method.

-Some background of domain knowledge is missing, such as the explanation of Fisher efficiency and Empirical Fisher Information Matrix, thus making part of the paper hard to follow.

**Questions:**

Most are mentioned in Weakness.

Another suggestion:

-Mistake of Bold fonts (STS-B, Natural Galore rank=4)

---

### Official Review · Reviewer_o6g5 · 2024-11-01

**Soundness:** 3
**Presentation:** 3
**Contribution:** 2
**Rating:** 5
**Confidence:** 4

**Summary:**

The paper introduces a novel memory efficient optimization scheme called Natural Galore. It draws inspiration from Galore, which significantly reduces the memory load caused by gradients and optimizer states during the training, and natural gradient descent, which belongs to approximate second-order methods and thus has a faster convergence than a first order method. Natural Galore is empirically evaluated on a set of NLP tasks by training neural networks of different sizes showing the competitiveness of the method with respect to the state-of-the-art algorithms.

**Strengths:**

The authors have well motivated their algorithm both from the memory efficiency and optimization perspectives.
To perform efficiently the update based on Fisher Information Matrix, they have introduced an efficient technique to compute the inverse of it, which is discussed in Section 2.4.

To show the effectiveness of Natural Galore, they have evaluated their method on a wide set of NLP tasks. The improvement on the function calling task in advanced agentic systems looks especially interesting.

Overall, the paper is well written and easy to follow.

**Weaknesses:**

There are no theoretical guarantees provided in the paper, only an extensive discussion on why this algorithm should work. At the same time, the experiments have been only executed on the NLP tasks and thus this empirical evaluation may not suffice to conclude that the method is competitive with other optimization algorithms on other tasks, for example, in computer vision. Moreover, there are some unclear moments related to evaluation and comparison methodology (see questions below).

It can be also argued that the ideas proposed in the paper lack novelty, as both low-rank based methods and natural gradient descent methods have already existed for quite awhile. So without significant evidence of superiority of Natural Galore over other low-rank based methods, the contribution is somewhat incremental.

Minor remarks:

line 231: a missing citation

As you have done a small survey on the memory efficient training in the introduction of the paper, consider to mention activation offloading

Rhu, M., Gimelshein, N., Clemons, J., Zulfiqar, A., and Keckler, S. W. (2016). vDNN: Virtualized Deep Neural Networks for Scalable, MemoryEfficient Neural Network Design

and combined activation checkpointing and offloading:

Beaumont, O., Eyraud-Dubois, L., and Shilova, A. (2021a). Efficient Combination of Rematerialization and Offloading for Training DNNs.

**Questions:**

- Line 230: Can you please elaborate what are past batches here?
- Lines 328-330: What is a typical value of $s$ in $O(s^2)$?
- Lines 233 and 333: It seems like two different formulations of optimization steps. Can you please clarify step-by-step how do you compute an update in Natural Galore? For example, do you use a momentum when computing an update? Is your update just a product of the inverse of fisher matrix and gradient?
 - (related to the previous question) What are optimizer states in Natural Galore, is it just a gradient and a pseudo Fisher matrix? If you estimate memory as in the paragraph between lines 040-047, how much bytes do you need to perform your update? And how does it compare to Galore?
- Line 402: Does it mean that each optimization method has the same lr? If yes, wouldn't it be fair to compare the algorithms by choosing the learning rate separately for each method as each algorithm may exhibit different training dynamics and thus benefit from different step-sizes?
- In Table 2, can you please provide an std as well, as for some training the improvement is only marginal and thus it is not clear if the result is statistically significant enough?
- Line 459: You mention that the best-performing model is selected, can you please elaborate on that? Do you mean that you train multiple models at the same time?
- In Section 3.3, why don't you compare with Galore for this task?

---

### Official Review · Reviewer_nn3u · 2024-11-04

**Soundness:** 2
**Presentation:** 2
**Contribution:** 2
**Rating:** 3
**Confidence:** 4

**Summary:**

The paper proposed Natural GaLore which is a modified version of the ICML 2024 Oral work "Gradient Low-Rank Projection" (GaLore) for improving the performance of pre-training and fine-tuning of LLMs. In particular, Natural GaLore introduces the second order regularizer information to the existing low-rank gradient estimate (of GaLore) with intention to better capture the optimization dynamics that was left wanting in GaLore due to low-rank approximation of the optimizer states. Natural Galore applies the inverse Empirical Fisher Information Matrix to low-rank gradients using Woodbury's Identity to represent the computationally expensive second order (quadratic) Hessian matrix term in the low-rank gradient descent.

**Strengths:**

1. The paper adds to the body of work (such as GaLore) focussing on optimizer/gradient efficient techniques for pre-training and fine-tuning LLMs, orthogonal to parameter efficient "fine-tuning" methods like LoRa/Q-Lora. Hence, this work helps bring memory-efficient pre-training (and fine-tuning) to LLMs. In addition, it helps paint the various related work in parallel and distributed training techniques

2. The work seeks to cover the performance gap between low-rank gradient i.e. GaLore and Full-rank pertaining and fine-tuning of LLMs by leveraging the second order term from Taylor series expansion of the update rule. It is well understood fact that second-order Hessian term (from double derivative of loss function) helps capture the curvature of the optimization landscape. The work finds a way to not ignore this term for sake of compensating for the performance loss from optimizer approximation in GaLore.

3. Natural GaLore maintains the significant memory-savings achieved by its underlying technique GaLore compared to LoRA - acting as drop-in replacement

**Weaknesses:**

1. The main contribution of the work is to improve performance of GaLore while maintaining the memory-efficiency. As a result, the presentation of the work seems to be incremental in its current form.

2. The performance results comparing Natural GaLore to LoRA do not hold much relevance as the significant improvements compared to LoRA was achieved by the root work i.e.  ICML 2024 GaLore. When comparing Natural Galore to its predecessor GaLore, which should be considered as the main point of comparison, the improvements are not as significant reported in Table 1 (Pretraining LlaMa, 3.53 vs 3.56, 3.22 vs 3.24,.. etc)  and Table 2 (Finetuning RoBERTa-Base).

3. It seems some reported results in Table 2 have inconsistencies (bold-faced) for the best performing methods for some tasks, eg:
- CoLa, Natural Galore 61.5 > 61.38 but LoRA is bold-faced,
- MRPC GaLore is better 92.25 > 92.10 than proposed Natural Galore , but latter is bold-faced. In addition, it is unclear why Natural galore performance with improved optimization dynamics degrades here compared to GaLore?

4. For results on TinyAgent models, GaLore performance has not been reported which could have helped our understanding of LoRA vs GaLore vs Natural Galore, and how much improvement Natural Galore adds to existing GaLore.

5. The presentation of the manuscript could be improved with explicit section/sub sections on Motivation, Related work, Contributions, Algorithm/Pseudocode of Natural Galore with "new" components on exisiting GaLore code, Table of comparison between LoRa, GaLore and Natural Galore (with check marks and crosses indicating how each of these are distinct along supporting fine-tuning, pretraining,  low-rank approximation on which part, memory cost, computation cost, etc)

**Questions:**

Q1: How significant are performance improvements of Natural GaLore (thereby its novelty) with respect to GaLore? Table 1: 3.53 vs 3.56, 3.22 vs 3.24, etc  and Table 2 improvements seems marginal thereby highlighting superiority of the existing GaLore compared to contributions of the newer version in Natural GaLore.

Q2: What is additional cost associated with Natural GaLore compared to GaLore for pre-training and fine-tuning tasks and its tradeoff to above minor performance improvements?

Q3: What is special about task CoLa in Table 2 that favors LoRA compared to GaLore and Natural GaLore?

Q4: How are average performance values calculated in Table 2 across tasks?

Q5: What is performance of GaLore on TinyAgent and how it compares to others in Table 3 ?

Q6: It would be helpful if Section 1: Introduction could be presented with motivation, related work, (itemized) contributions and a summary table comparing LoRa vs GaLore vs Natural Galore (with say check marks and crosses indicating how each of these are distinct along supporting fine-tuning, pretraining,  low-rank approximation on which part, quantifying memory cost and computation cost, etc)

Q7: It would help understand Natural Galore better if a pseudocode of GaLore and Natural Galore could be presented, and help quantify additional cost during training while maintaining memory cost.

Q8: Under PEFT paragraph in Introduction, please define m and n.

Q9: Line 231: (cite Fuji et al. paper) - A reference is missing.

Q10: Linespace missing between Line 365 and Line 366.

---

### Official Review · Reviewer_5AVo · 2024-11-05

**Soundness:** 2
**Presentation:** 2
**Contribution:** 2
**Rating:** 5
**Confidence:** 2

**Summary:**

This paper proposes Natural GaLore, which integrates natural gradient descent with GaLore for training, allowing GaLore to leverage missing second-order information in the optimization process to enhance performance. Specifically, Natural GaLore incorporates second-order information through the inverse Empirical Fisher Information Matrix (FIM) applied to low-rank gradients using Woodbury’s Identity.

**Strengths:**

- The integration of natural gradient descent with GaLore is an interesting and promising approach. It maintains the same memory usage as GaLore while achieving better performance, demonstrating the method’s effectiveness.

- The method is applied to a TinyLlama task as a case study, showcasing its practical utility.

**Weaknesses:**

The paper dedicates a large space to past works and background information, with only about one and a half pages (sections 2.3 and 2.4) covering the main technique. Even within these sections, it is difficult to distinguish the novel contributions from related work, as the content is intermingled.

Given that this is a training optimization method, the paper should include a convergence analysis. Additionally, a discussion on the generality and scalability of this method, such as how the performance curve behaves with larger steps, would be beneficial.

The paper should include ablation studies, particularly showing the training curve of the natural gradient descent. This would help in understanding the contributions of this paper.

For the TinyLlama results, the paper does not compare with the full-rank results for the 1.1B model, which would provide a clearer benchmark.

Besides memory savings, the authors are suggested to analyze the computational cost and discuss the ease of integrating the method into existing training frameworks.

The introduction contains extensive content on parallel and distributed training techniques, which seems unnecessary as these methods are orthogonal to the proposed method. The introduction should be focused on your research question and contributions.

**Questions:**

Could you add some discussion about the generality and scalability of this method?

---

### Note · Authors · 2025-01-06

**Comment:**

I would like to withdraw my submission and resubmit after redrafting at a future conference

**Withdrawal Confirmation:**

I have read and agree with the venue's withdrawal policy on behalf of myself and my co-authors.